# Extrachromosomal circular DNA and structural variants highlight genome instability in *Arabidopsis* epigenetic mutants

Panpan Zhang[1,2,3], Assane Mbodj[1,2], Abirami Soundiramourtty[2,4], Christel Llauro[2,5], Alain Ghesquière[6], Mathieu Ingouff [6], R. Keith Slotkin [7,8], Frédéric Pontvianne [5], Marco Catoni [9] & Marie Mirouze [1,2] ✉

Abundant extrachromosomal circular DNA (eccDNA) is associated with transposable element (TE) activity. However, how the eccDNA compartment is controlled by epigenetic regulations and what is its impact on the genome is understudied. Here, using long reads, we sequence both the eccDNA compartment and the genome of Arabidopsis thaliana mutant plants affected in DNA methylation and post-transcriptional gene silencing. We detect a high load of TE-derived eccDNA with truncated and chimeric forms. On the genomic side, on top of truncated and full length TE neo-insertions, we detect complex structural variations (SVs) notably at a disease resistance cluster being a natural hotspot of SV. Finally, we serendipitously identify large tandem duplications in hypomethylated plants, suggesting that SVs could have been overlooked in epigenetic mutants. We propose that a high eccDNA load may alter DNA repair pathways leading to genome instability and the accumulation of SVs, at least in plants.

Extrachromosomal circular DNA (eccDNA) has been described for decades in eukaryote cells including fungi, Drosophila, nematodes, plants and mammals[1–4]. Only recently this fraction of the cell genetic material has gained attention thanks to the development of specific eccDNA-sequencing approaches allowing its characterization, such as mobilome-seq[5,6], CIDER-seq[7] and circle-seq[8]. EccDNA can originate from spurious homologous recombination between tandem copies (for example ribosomal DNA or telomeric DNA) or micro-homologies[9]. EccDNA can also be formed from circularization of extrachromosomal linear DNA of active transposable elements (TEs)[5,10] through homologous recombination (HR) or non-homologous end-joining (NHEJ). The roles of eccDNA are just being unraveled. For example the presence of eccDNA is associated with senescence in yeast and with apoptosis in mammal cells[11,12]. In stressful environmental conditions, eccDNA can provide the cells with an advantage. For instance in yeast, eccDNA arise in nitrogen limiting conditions and contributes to stress adaptation[13]. In cancer cells, abundant eccDNAs originating from chromothripsis (a massive event of genome rearrangement) can lead to oncogene amplification and contribute to tumor evolution[14–16]. In weeds, eccDNA encoded genes can contribute to herbicide resistance[17], demonstrating the adaptive role of the eccDNA compartment in plants exposed to strong selective pressure[18]. Recently, *PopRice*, an abundant TE-derived eccDNA in rice endosperm[5] was shown to act as a sponge for a transcription factor during germination[19]. The role of eccDNA upon their potential re-integration in the genome has been characterized in cancer cells[16] but the interplay between eccDNA and genome stability is poorly described in other biological systems. In yeast and cattle, early works suggested that

[1]Institut de Recherche pour le Développement (IRD), Laboratory of Plant Genome and Development, Perpignan, France. [2]EMR269 MANGO (CNRS/IRD/UPVD), Laboratory of Plant Genome and Development, Perpignan, France. [3]University of Montpellier, Montpellier, France. [4]University of Perpignan, Perpignan, France. [5]Centre National de la Recherche Scientifique (CNRS), Laboratory of Plant Genome and Development, Perpignan, France. [6]DIADE, University of Montpellier, IRD, CIRAD, Montpellier, France. [7]Donald Danforth Plant Science Center, St. Louis, MO 63132, USA. [8]Division of Biological Sciences, University of Missouri, Columbia, MO 65211, USA. [9]School of Biosciences, University of Birmingham, Birmingham B15 2TT, UK. ✉e-mail: marie.mirouze@ird.fr

eccDNAs could be linked to genomic structural variations (SVs)[20,21]. Other indirect evidence links the presence of highly active TEs and SVs in maize[22]. Indeed SVs are enriched with repeated DNA including TEs[23–25], as exemplified for NLR genes in pepper retrogenes[26] or for the *sun* locus in tomato[27]. More recently, large scale studies of SVs in tomato identified DNA repeats in around 80% of SVs[28]. To date, most TE-mediated SVs have been identified in natural variants, and the mechanism underlying these TE-mediated SVs is not yet clear, notably for the role of TE-eccDNA. The lack of ongoing eccDNA accumulation in model plants has prevented a comprehensive analysis of the impact of eccDNA on genomic SVs.

In plants, TEs are controlled at different steps in their life cycle by a combination of epigenetic regulations involving notably DNA methylation (for gene silencing) and post-transcriptional gene silencing (PTGS)[29]. Mutant plants affected in one or a combination of these pathways have a high level of TE transcription[30,31]. DNA methylation is maintained directly by methyltransferases and by the RNA-directed DNA methylation pathway (RdDM). It is also indirectly maintained by the chromatin remodeler DDM1 (Decrease DNA Methylation 1) involved in the deposition of the heterochromatic histone variant H2A.W at full-length TEs[32]. DDM1 belongs to DHL chromatin remodelers, an acronym for a family also comprising human HELLS (Human helicase lymphoid specific) and mice Lsh (Lymphoid specific helicase)[33]. Recent studies suggest a role of histone variants in DNA repair[34]. For instance, HELLS plays a role in HR repair of heterochromatic breaks[35] and Lsh promotes DNA repair in mice[36]. HELLS and Lsh are expressed in testis and in lymphoid cells, where they participate in V(D)J recombination. In *Arabidopsis thaliana*, DDM1 ensures HR repair of double strand breaks (DSBs)[37]. Loss of DDM1 is associated with a complete change in the epigenome, notably through changes in DNA methylation, histone methylation at H3K9, and H2A.W [33]. Because of its multiple roles, we wondered whether loss of DDM1 could have an impact on the accumulation of eccDNA in plants.

Here, using a reverse genetics approach, we can perturb the regulation of eccDNA and directly address the role of eccDNA on genome stability in *A. thaliana* mutant plants. In order to further the load of TE-derived eccDNA in *A. thaliana* (hereafter TE-eccDNA), we use a combination of mutant plants affected in *DDM1*, *RDR6* (involved in PTGS) and *NRPD1* (the largest catalytic subunit of the *POL IV* holoenzyme that participates in RdDM). We perform mobilome-seq and long read genome resequencing to characterize both the eccDNA and genome contents and analyze the impact of eccDNA on genomic stability. Thanks to long read mobilome-seq, we show that TE-eccDNA can be truncated or chimeric. By analyzing the genomic content of these mutants plants, we uncover examples of SVs suggesting a high level of genome instability in these genetic mutant backgrounds.

## Results

### Detection of eccDNAs and TE-mediated SVs in epigenetic mutants

To characterize the eccDNA repertoire of the wild type and epigenetic mutants (*ddm1*, *polIV*, *rdr6*, *ddm1 polIV*, *ddm1 rdr6*, and *ddm1 polIV rdr6*), a series of 21 mobilome-seq experiments were conducted (Fig. 1a). Briefly, after removal of genomic linear DNA with exonuclease V, the circular DNA molecules were amplified by random rolling circle and sequenced using Illumina or ONT platforms (Supplementary Fig. 1). Additionally, thirteen whole genome resequencing datasets using ONT long reads were produced from these samples, to investigate SVs (Fig. 1a). The most abundant class of TE-eccDNA was identified as corresponding to the long terminal repeat (LTR) retrotransposon *EVD*, a TE previously observed as eccDNA in hypomethylated mutants[38]. *EVD*-eccDNA were present in all libraries from *ddm1* mutant combinations

mutants, but not in *polIV* or *rdr6* single mutants (Fig. 1b). The sequencing coverage at the *EVD/AT5G17125* locus reached up to 37,260X in the triple mutant, compared to 6X in the WT. The second enriched TE-eccDNA corresponds to the DNA transposon of the Mutator family *VANDAL21* (Supplementary Fig. 2). The sequencing coverage at the *VANDAL21/AT2TE42810* locus reached up to 2,080X in the triple mutant, compared to 6X in the WT. At both *EVD* (Fig. 1c) and *VANDAL21* (Supplementary Fig. 2) loci the reads cover the entire elements, confirming that these loci lead to eccDNA formation. In order to capture the full picture of the circular structures, we performed mobilome-seq using ONT reads on the WT and *ddm1 polIV rdr6* triple mutant plants. Thanks to eccDNA amplification with random rolling circle, the ONT sequencing reads contain tandem repeated units of the eccDNA template (Supplementary Fig. 1), allowing us to decipher the real structure of eccDNAs. Compared to Illumina mobilome-seq, ONT mobilome-seq data yielded a higher and more homogenous sequencing depth across both *EVD* (Fig. 1c) and *VANDAL21* (Supplementary Fig. 2), showing clearer boundaries. The accumulation of reads corresponding to both TE families was specific to the triple mutant and consistent in the two biological replicates (Supplementary Fig. 3). In contrast, reads corresponding to chloroplast and mitochondrial circular genomes and to endogenous rDNA eccDNAs (positive controls also amplified in our procedure) were detected in all samples (Supplementary Fig. 3). Altogether these data confirm the formation of *bona fida* eccDNAs from both *EVD* and *VANDAL21* TE families. Among the 318 TE families that we analyzed, these two were the only ones enriched in both biological replicates of the triple mutant compared to the WT (Fig. 1). For example we found evidence of eccDNA for *ATCOPIA51* and *ATCOPIA52/SYSIPHUS* only in one biological replicate.

To investigate the activity of corresponding TEs, we analyzed genomic SVs mediated by *EVD* and *VANDAL21*, and detected up to 73 and 6 new insertions for *EVD* and *VANDAL21*, respectively, in a pool of 3 *ddm1 polIV rdr6* mutant plants (Fig. 1d). In the single *ddm1* individual plant we detected 26 *EVD* insertions, whereas in individual *ddm1 polIV* we detected 56 and 1 insertions for *EVD* and *VANDAL21*, respectively (Fig. 1d). Intriguingly in *ddm1 rdr6* we did not detect any *EVD* insertion. Additionally, in *ddm1 polIV* and *ddm1 polIV rdr6* mutants we detected insertions of two DNA transposons (*ATENSPM3* and *ATMU5*) and the retrotransposon *ATCOPIA21*, not detected as eccDNA and that could be active in earlier generations or at a different developmental stage. *Copia* retrotransposons (*EVD* and *ATCOPIA21*) inserted preferentially within exons, *VANDAL21* mainly targeted the 5′ UTRs of active genes, and *ATENSPM3* insertion sites were more widely distributed next to genes (Fig. 1e).

### Truncated TEs are present as both eccDNA and new insertions loci

*Ty1/Copia* retrotransposons comprise two LTRs surrounding an open reading frame that encodes a GAG (Group-specific Antigen) and a POL (polyprotein), which is subsequently cleaved into four active functional domains, namely AP (aspartic protease), IN (integrase), RT (reverse transcriptase) and RH (RNase H) (Fig. 2a). In addition to eccDNAs corresponding to expected full-length *EVD* with one or 2 LTRs (Fig. 2b), we identified eccDNAs with partial structures: loss of GAG domain (13%), loss of IN and RT domains (19%), and loss of LTR (20%). These truncated *EVD*-eccDNAs account for 52% of all *EVD*-eccDNAs, while full-length *EVD*-eccDNAs represent 48% of all *EVD*-eccDNAs (Fig. 2b). No hotspot for truncation could be detected (Supplementary Fig. 4), despite a very high coverage with ONT reads (up to 34,695X, Fig. 1c).

Given the high frequency of truncated eccDNAs we analyzed their potential impact on genomic SVs. In this aim we selected *EVD*

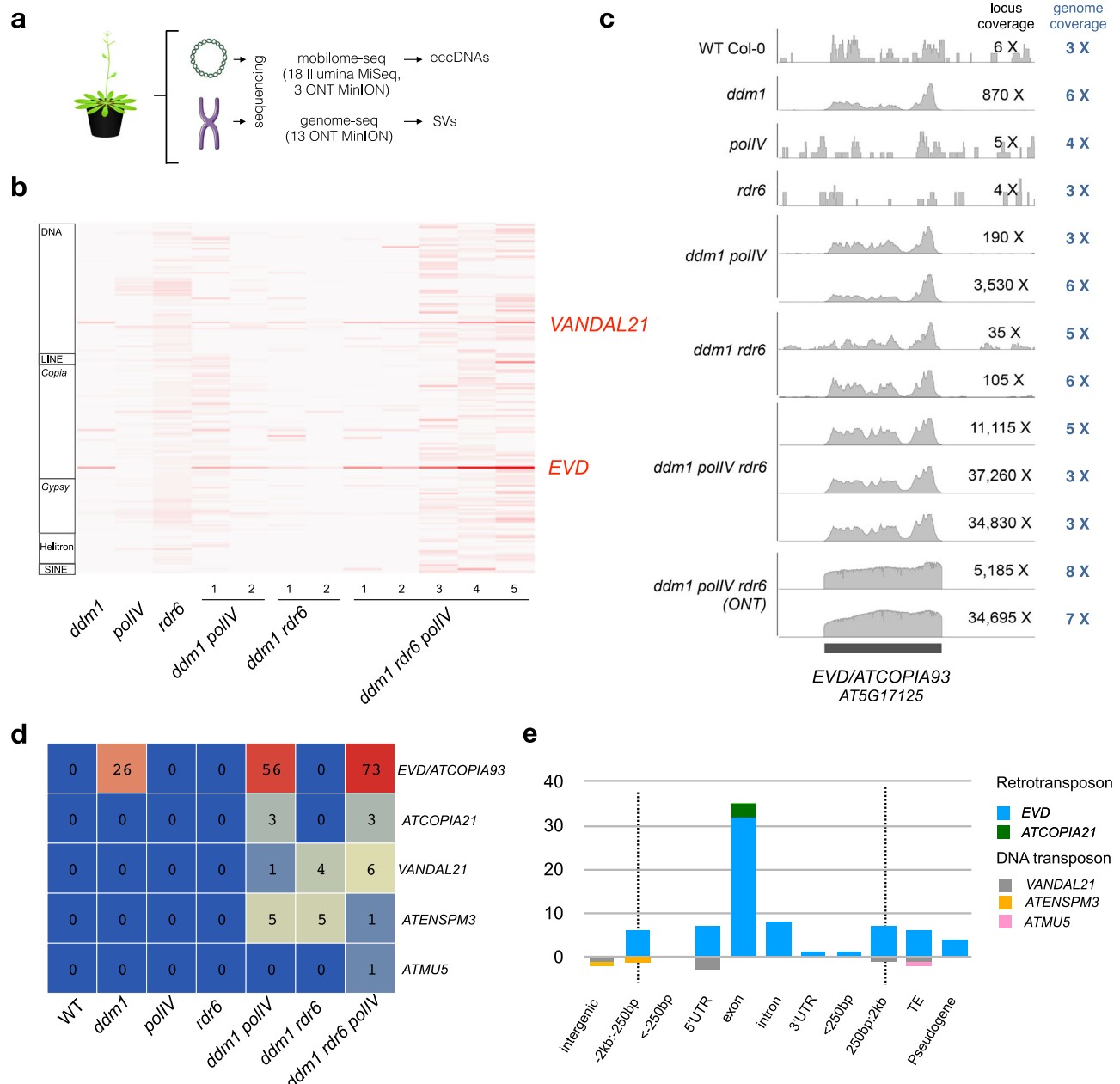

**Fig. 1 | High eccDNA load in *Arabidopsis thaliana* epigenetic mutants is associated with new TE insertions in the genome. a** Scheme depicting the experimental design: from Arabidopsis plants both linear and circular DNA were extracted and sequenced using Illumina and Nanopore (ONT) platforms in order to characterize the eccDNA repertoire and the genomic SVs. **b** Heat map showing the abundance of TE-eccDNAs detected for 318 *Arabidopsis* TE families in *ddm1*, *polIV*, *rdr6*, *ddm1 polIV*, *ddm1 rdr6* and *ddm1 rdr6 polIV* triple mutant plants using Illumina mobilome-seq. The number of eccDNA is normalized by the number of mapped reads per million per library and compared to the WT. **c** Mobilome-seq data using Illumina and Nanopore sequencing (ONT) showing the coverage at an *EVD* locus (Chr5:5630538-5634764). ONT reads result in clearer boundaries and uniform coverage. **d** Number of insertions for 7 mobilized TE families in the genome of *ddm1*, *ddm1 rdr6*, *ddm1 polIV* and *ddm1 rdr6 polIV* mutants. Note that single plants were sequenced except for the triple mutant where a pool of 5 plants was used, because of the dwarf phenotype. **e** Target site preferences for different TE families in *ddm1 rdr6 polIV* mutant plants. Source data are provided as a Source Data file.

containing reads in our ONT genome resequencing datasets for WT, *ddm1*, *ddm1 polIV*, and *ddm1 polIV rdr6* mutants. We excluded single *polIV* and *rdr6* mutants and *ddm1 rdr6* double mutants as there was no new detected *EVD* insertion in these mutant backgrounds. New insertions corresponding to truncated retrotransposon structures were detected at distinct loci in mutants (Fig. 2c, d). We also observed a truncated insertion for *ATCOPIA21* (Supplementary Fig. 5). Two of these truncated insertions contain clear target site duplications (TSDs). Of note, all truncated insertions possess two LTRs (Fig. 2c, d) suggesting that only a fraction of truncated TEs are linked to new insertion events.

## Both eccDNAs and genomic SVs contain chimeric TEs

All eccDNAs described so far consisted exclusively of TE-derived DNA. In contrast, in *ddm1 polIV rdr6* plants we observed a striking example of chimeric eccDNA, defined as a circle originating from different genomic loci. In this eccDNA, supported by 7 copies in a single long read (Fig. 3a), a partial *EVD* fragment and a portion of the *AT5G66440* gene (encoding a tRNA methyltransferase) had fused. *EVD* containing reads were extracted from ONT mobilome-seq datasets for the triple mutant plants and mapped to the reference genome. The mapping profile confirmed the presence of chimeric reads at the *AT5G66440* locus (Fig. 3b). Additionally,

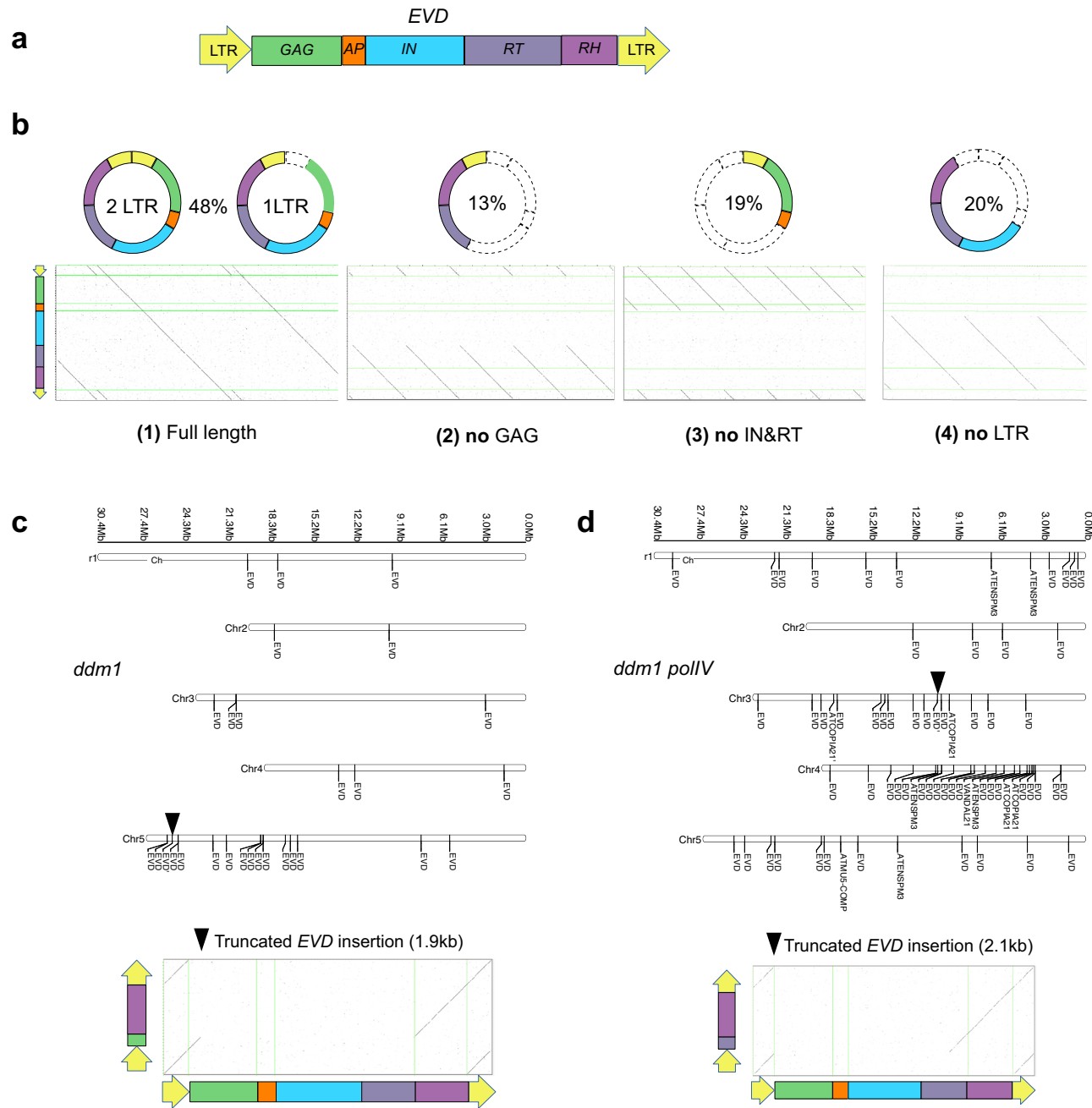

**Fig. 2 | Truncated *EVD* are detected as eccDNA and as new integrated genomic copies in *A. thaliana* epigenetic mutants. a** A schematic view of *EVD* structure showing the main domains: LTR (yellow arrow), GAG (green), aspartic protease (AP, orange), integrate (IN, cyan), reverse transcriptase (RT, purple) and RNase H (RH, dark orchid). **b** *EVD*-eccDNAs were classified into 4 categories. For each category an example is shown as a dot plot for an individual representative read. The categories are: (1) Full length: a single read contains more than 2 full-length copies of *EVD* (1 or 2 LTRs circles); (2) no GAG: a read contains more than 5 truncated *EVD* without GAG, AP and IN domains; (3) no IN&RT: a read contains more than 5 truncated *EVD* without IN and RT domains; (4) no LTR: a read contains 3 truncated *EVD* without LTR, GAG and AP. Localization of new TE genomic insertions in *ddm1* (**c** *EVD* insertions) and *ddm1 polIV* (**d** insertions of *EVD*, *ATENSPM3*, *COPIA21*, *VANDAL21* and *ATMU5*) mutants. Truncated *EVD* insertions are shown as black triangles. The bottom part of each panel shows a dot plot between the insertion and the full-length *EVD* for individual genomic ONT reads. Source data are provided as a Source Data file.

we detected an *EVD* insertion at the corresponding locus (*AT5G66440*) in the genome of the same plants (Fig. 3c), flanked by a 5 bp TSD, a signature of integrase-mediated insertion (Fig. 3d). This finding suggests that the chimeric eccDNA might originate from the new insertion of *EVD* at this locus.

To investigate whether this chimerism was specific to the *EVD* family or could be detected for other TE families, we extracted *ATCOPIA21* long reads from two replicates of *ddm1 rdr6 polIV* ONT genomic data. We noticed that these *ATCOPIA21*-containing reads

displayed complex re-arrangements when mapped to the reference genome. They were divided into three segments mapping to distinct genomic regions, and here-after referred to as "3-hit" reads. One "3-hit" read started from the *AT4G16950* gene encoding the pathogen response gene *RPP5* (for *RECOGNITION of PERONOSPORA PARASITICA 5*), then spanned *ATCOPIA21*, and ended at the *AT4G16970* gene encoding a CRK19 (CYSTEIN-RICH RECEPTOR-LIKE PROTEIN KINASE 19) located 3 Mb away from the *RPP5* cluster on the reference genome (Supplementary Fig. 6). This was confirmed by mapping the

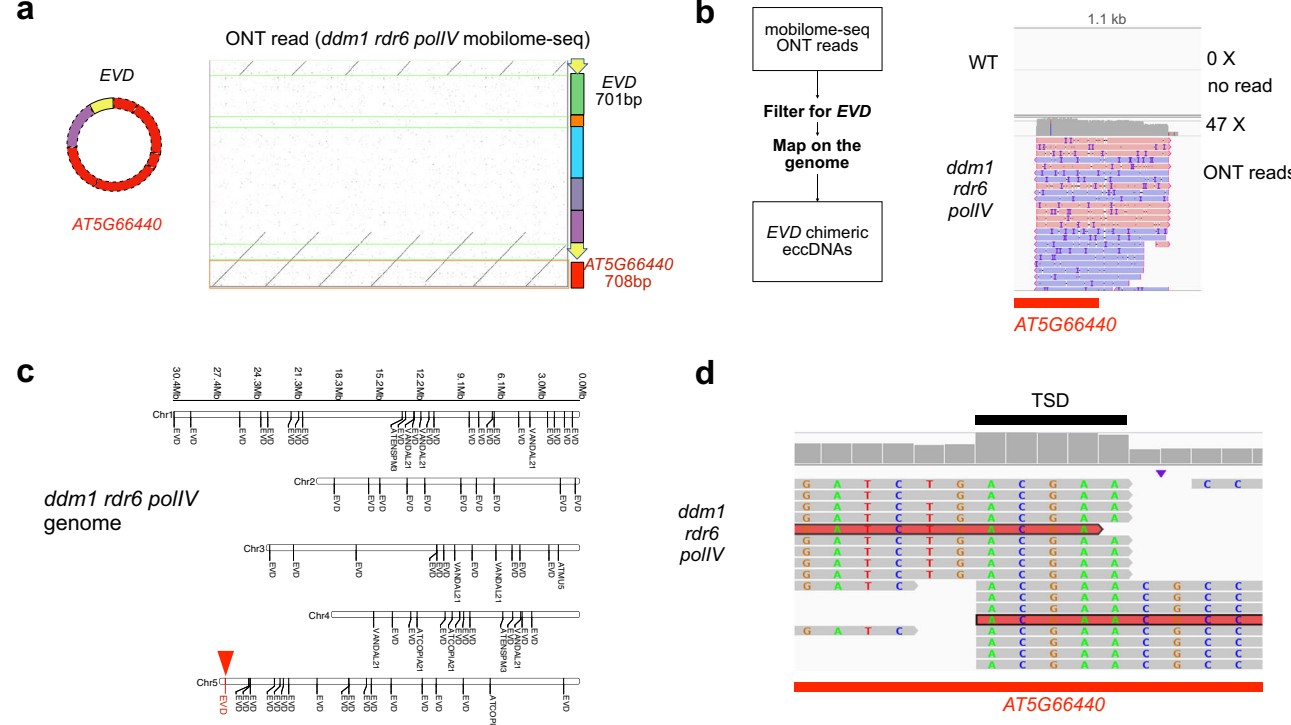

**Fig. 3 | Chimeric eccDNA containing a truncated *EVD*-gene fusion corresponds to a chimeric genomic insertion in *A. thaliana ddm1 rdr6 polIV* mutant. a** Dot plot between an ONT read from the *ddm1 rdr6 polIV* mutant mobilome-seq and corresponding *EVD* (color code as in Fig.2) and *AT5G66440* sequence (*red*). The deduced chimeric circle is depicted on the left, consisting of a truncated *EVD* fragment (701 bp) fused to a truncated *AT5G66440* gene fragment (708 bp).

**b** eccDNA-seq ONT reads containing *EVD* were selected and mapped to the reference genome and shown at the *AT5G66440* locus. **c** New TE insertions detected in the *ddm1 rdr6 polIV* mutant genome. A new copy of *EVD* is detected at the *AT5G66440* locus (red triangle). **d** The *EVD* insertion at *AT5G66440* created a 5 bp target site duplication (TSD, black line), visible thanks to read mapping locations ending or starting at the same position on the reference.

*ATCOPIA21* containing ONT reads on the reference genome at both *RPP5* and *CRK19* loci. The insertion of *ATCOPIA21* at the *CRK19* gene created a 9 bp TSD (TATAGTAGC). This TSD seems unusually long[39]. but suggests a proper integration event. This structural remodeling resulted in *RPP5* gene duplication close to the *CRK19* gene in the *ddm1 rdr6 polIV* triple mutant (Supplementary Fig. 6).

The *RPP5* locus is highly repetitive (Supplementary Fig. 7). The detection of chimeric duplicated *RPP5* locus in *ddm1 rdr6 polIV* prompted us to analyze the stability at the endogenous *RPP5* locus in this mutant background. While we did not detect any SV neither in Col-0 nor in *ddm1* at this locus, we detected a 13.3 kb deletion in the triple mutant, in one biological replicate (Supplementary Fig. 7). This replicate comes from a pool of 3 plants, suggesting that one plant could be heterozygous for the deletion. *ddm1* mutants have hypomethylated TEs. In order to test a different line showing hypomethylated and active TEs, we analyzed *met1*-derived epiRILs genomes[40,41]. When compared to the Col-0 control, 9 out of 21 lines analyzed displayed a lack of coverage at the *RPP5* locus, compatible with a recombination event between the two flanking genes *AT4G16940* and *AT4G16960* (Supplementary Fig. 8). This result suggests that, in independent conditions of hypomethylation, this locus displays genomic instability.

**Large SVs in hypomethylated plants beyond eccDNA**
The prevalence of detected SVs in the epigenetic mutants prompted us to characterize all SVs in these mutants, beyond TEs producing TE-eccDNAs. We detected a 55 kb (Chr1:5,548,395-5,603,615) and a 56 kb duplication (Chr2:231,518-287,731) in two *ddm1* siblings, respectively (Fig. 4a). The first duplication starts at the 5'UTR of *AT1G16220* and ends within the *AT1G16390* exon (Fig. 4b). We analyzed long reads

crossing the junction of the two tandem copies and confirmed the 55 kb duplication by dot plots (Fig. 4c). We did not detect SNPs in the two tandem copies, but the second copy is slightly hypomethylated at the left breakpoint compared to the first one (Supplementary Fig. 9). This suggests that the two tandem copies are recent, differing only in their epigenetic patterns. The second duplication is 56 kb long, starting in the exon 2 of the *AT2G01510* gene and ending between two DNA transposons (*AT2TE01165* and *AT2TE01180*) (Fig. 4d). A third 34 kb long duplication was identified in *ddm1 polIV* in a repeated region (Supplementary Fig. 10).

We did not identify eccDNAs corresponding to these large segmental duplications suggesting that these eccDNA forms could be transient or that these SVs could be initiated though other pathways. We reasoned that the accumulation of eccDNA might affect the DNA repair pathway. Large tandem duplications have been observed in *Caenorhabditis elegans* mutant affected in DNA repair, together with small deletions[42]. To evaluate whether *ddm1* mutant phenocopies this DNA repair mutant, we analyzed small deletions in single plants of *ddm1* and *ddm1 polIV* and found 85 and 51 deletions absent in the WT, respectively (Fig. 4).

## Discussion
We investigated both eccDNA content and SVs in *A. thaliana* epigenetic mutants in order to characterize the eccDNA repertoire and document whether it can impact genomic SVs (Fig. 5). In WT plants the formation of TE-eccDNAs is prevented by DNA methylation pathways and PTGS. In *ddm1 rdr6 polIV* mutant plants, we detected a heavy load of eccDNAs, most of them originating from *EVD*, an LTR retrotransposon active in hypomethylated plants[38,43] and *VANDAL21*, a DNA transposon active in *ddm1*[43]. Long read sequencing of eccDNA revealed that around half of

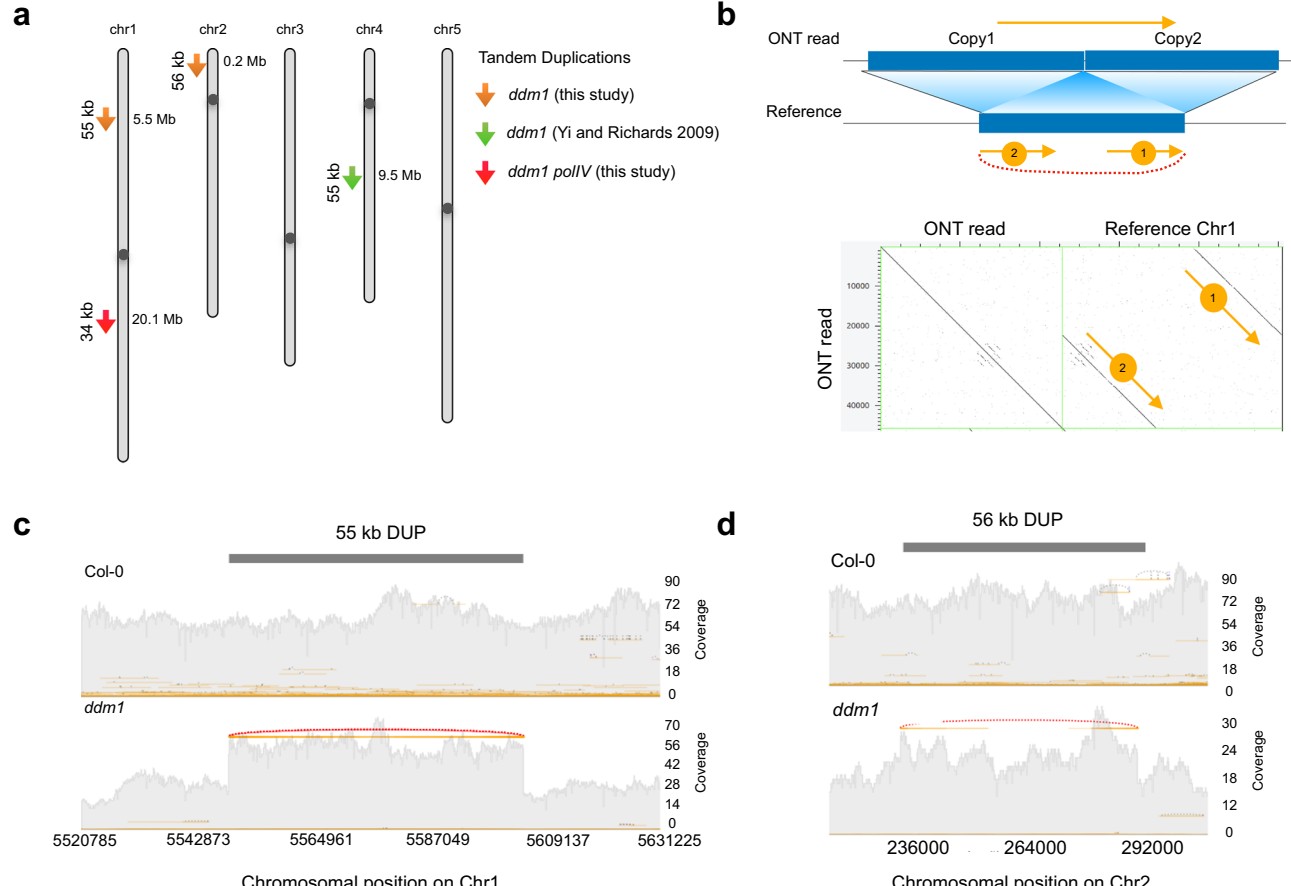

**Fig. 4 | Detection of large SVs in *ddm1* and *ddm1 polIV* mutant plants. a** Location of large segmental duplications reported in *ddm1* and *ddm1 polIV* mutants. Duplications (arrows) sizes and locations are indicated. **b** Scheme indicating how the ONT reads (orange arrows) spanning the junction of a tandem duplication will be split and aligned on the reference. As an example a dot plot showing a raw 46 kb-long ONT read (b8A44112-cc27-49b3-9b48-1807A273fd3d) versus the reference at the duplicated region on Chr1 (Chr1:5548395-5603615) is shown. Examples of read depth of the *A. thaliana* wild type Col-0 and *ddm1* ONT genomic reads aligned to the TAIR10 reference genome on chromosome 1 (**c** 55 kb duplication) and 2 (**d** 56 kb duplication). Dotted red lines and gray rectangle highlight the duplication as detected by Sniffles. Source data are provided as a Source Data file.

*EVD*-eccDNAs were composed of truncated *EVD* in *ddm1 rdr6 polIV*. Truncated insertions of *EVD* were detected in the corresponding *ddm1 rdr6 polIV* genome, but also in single *ddm1* mutant plants, suggesting that the *ddm1* mutant background itself is sufficient to promote these truncated TE insertions. This points to a mechanism of TE internal deletion and formation of non-autonomous elements. Truncated insertions have been described earlier[44–47] however it is not yet clear how these insertions occur. Future investigations will be needed to characterize whether this phenomenon occurs (i) early during TE life-cycle in the VLP, as suggested by reshuffling for *Onsen*[48,49] or *ATCO-PIA52/SISYPHUS*[50], (ii) post-integration in the genome or (iii) if truncated eccDNA would be able to re-insert into the genome. TE-eccDNAs are generally considered as a dead-end, not being able to re-integrate into the genome[51]. However, in yeast, cattle and cancer cells, integration of eccDNAs has been suggested[20,21,52–54].

While mobilome-seq data suggested that *EVD* and *VANDAL21* were active in the *ddm1* mutant combinations, genome resequencing using long reads revealed insertions of TEs from additional families: *ATCO-PIA21, ATENSPM3* and *ATMU5*. Several hypotheses could explain why eccDNA corresponding to these 3 families could not be detected. These eccDNAs could be expressed in a tissue specific manner, or with a very limited short life, for yet unknown reasons, limiting our power to detect them. For instance, concerning class II elements, *Mu* and *Ds* elements preferentially insert in genes expressed in meristematic enriched tissues[55]. Future experiments of targeted mobilome-seq

could help addressing this issue. The activity of TE families might also differ between plants or lines of the same genotype, as shown before for *VANDAL21*, *CACTA*, *ATCOPIA13* or *ATGP3* for instance[43]. This could also explain why we did not detect new insertions of *EVD* in *ddm1 rdr6*, contrary to a previous report[50], and despite the presence of *EVD*-eccDNA.

On top of expected TE neo-insertions, we uncovered complex SVs in *ddm1* mutant combinations. For example we detected chimeric genic reads consisting of DNA sequences originating from *ATCOPIA21* and two genes fragments: *RPP5* and *CRK19*, revealing a partial duplication of the *RPP5* locus. This locus is located within a resistance (R) gene cluster that plays an important role in the innate immune response to pathogens in *A. thaliana*[56]. Clusters of R genes are critical for plant resistance in the evolutionary arms race with pathogens, and these regions are enriched for TEs[57]. Intriguingly we observed a large deletion at the *RPP5* locus both in one biological replicate of *ddm1 rdr6 polIV* mutants and in *met1*-derived epiRIL population suggesting that genetic instability could be a direct or indirect consequence of DNA hypomethylation at this locus. This observation is consistent with the lack of *RPP5* expression previously reported in *met1* epiRILs[58]. Some years ago a 55 kb duplication at this locus (named *bal*) had been reported and associated with a dwarf plant phenotype[59]. A parsimonious explanation could be that TE instability can lead to genomic SVs at this locus. It is also possible that TEs preferentially insert in regions of instability while SVs are caused by changes in DNA methylation

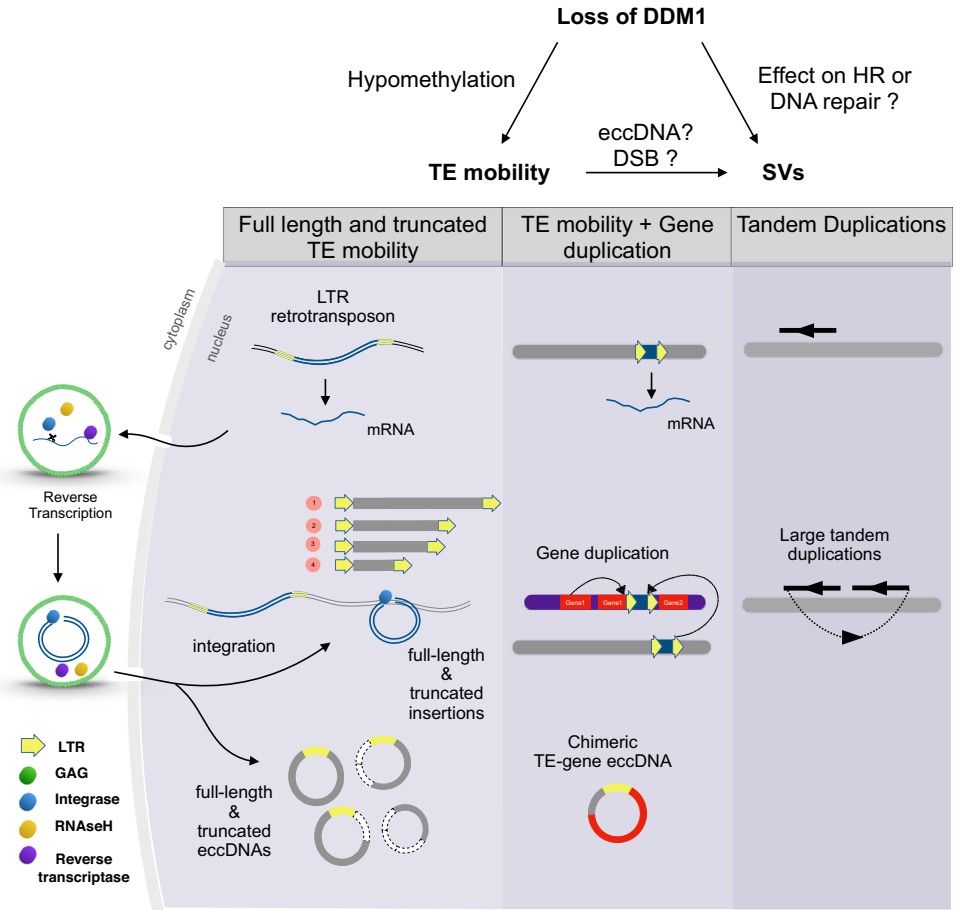

**Fig. 5 | Overall genomic instability detected in this study in *A. thaliana* epigenetic mutants.** Loss of DDM1 could have an impact on genome stability either through an unknown direct effect on DNA repair and homologous recombination (HR) or through its indirect effect on TE mobility triggering high eccDNA level and double strand breaks (DSBs).

regardless of TE insertion. The role of TEs in remodeling the genome is reminiscent of the recent discovery of LINE1-induced SVs in human cancer genomes[60]. More systematic analyses of individual plants with TE bursts and associated eccDNAs will allow addressing the extent of TE-driven structural remodeling.

Beyond chimeric or truncated TE insertions linked to eccDNA we have identified large tandem duplications and small deletions in single plants of *ddm1* and *ddm1 polIV*, not associated with eccDNA, suggesting an indirect link between the presence of eccDNAs and these SVs. Large tandem duplications have also been reported in lines derived from a *fas2* mutant[61]. Both loss of *DDM1* or *FAS2* correlate with higher rates of meiotic[62] or homologous recombination[63], respectively, showing that DNA repair pathways are affected in these mutants. Interestingly we found that the pattern of SVs in *ddm1* and *ddm1 polIV* (large tandem duplications and small deletions) phenocopies the BRCA1-associated SVs found in a *C. elegans* model[42]. During their life cycle, proliferating TEs produce linear extrachromosomal DNA that presents double strand breaks (DSBs) and eventually get circularized presumably though homologous recombination or non-homologous end-joining to produce eccDNA. In hypomethylated plants, the high load of eccDNA coupled with active integration of TEs (through integrase or transposase) could result in an excess of DSBs in the cell, overloading the repair machinery (Fig. 5). We propose that this could trigger alternative pathways for DNA repair, such as the recently described TMEJ involving the polymerase theta[64], leading to genomic SVs. Of note, while this study was in press, a publication reported the role of DNA repair pathways in eccDNA formation in Drosophila[65].

By sequencing both the mobilomes and genomes of *ddm1*, *ddm1 polIV* and *ddm1 rdr6 polIV* plants we detected a high eccDNA load associated with genome instability. We speculate that the abundance of eccDNA in *ddm1* mutant combinations triggers alternative DNA repair pathways leading to SVs and mimicking mutants in DNA repair. During evolution, TE-derived eccDNAs could contribute to hotspots of SVs and participate to the rapid evolution of disease resistance clusters in natural plant populations. Analyses in other biological models would allow testing the hypothesis that TEs and/or TE-eccDNA contribute to a « two speed » genome evolution mechanism by contributing to instability at genomic hotspots.

## Methods
### Plant material and growth conditions
Seeds from *A. thaliana* WT ecotype Columbia-0 and mutants were sown on soil, stratified for 2 days at 4 °C and grown in a growth chamber at 21 °C under long day conditions (16 h light). The mutant alleles used in this study were *ddm1-2*[66] first and six generation mutants, *rdr6-15* (SAIL_617) and *polIV/nrpd1-3*[67]. The mutant combinations have been previously described[68]. For eccDNA analyses, leaf material was harvested from a pool of 12 plants after 3 weeks. For ONT genome resequencing, aerial part from individual plants was harvested after 3 weeks, except for the dwarf *ddm1 polIV rdr6* mutant plants for which 3 siblings had to be pooled for DNA extraction. We performed at least two biological replicates for all DNA extractions, except for the *polIV* and *rdr6* single mutants where one replicate was used, as these single mutants have no reported evidence of active TEs.

## DNA extraction

Seedlings were collected into one tube immediately snap-frozen in liquid nitrogen and stored at −80 °C until DNA extraction, in duplicate or triplicate. Total DNA was extracted using the CTAB approach[5]. Total DNA quantity was measured with a Qubit Fluorometer (Thermo Fisher Scientific). Genomic DNA was quality controlled using Nanodrop measures, Qubit quantifications and visualization on a 1% agarose gel electrophoresis.

## eccDNA enrichment

eccDNA enriched was performed using the Phi29 enzyme[5]. Briefly, genomic DNA (2 µg) was first cleaned up using the PCR Cleanup kit (Qiagen) to remove large genomic fragments. The DNA on the column was eluted with 30 µL elution buffer. DNA concentration was measured with a Qubit fluorimeter. In order to eliminate linear DNA fragments, 25 µL of purified DNA was digested with the Plasmid-Safe ATP-Dependent DNase (Epicenter) according to the manufacturer's instructions. Following digestion, eccDNA was precipitated with 0.1 volume of 3 M sodium acetate (pH 5.2), 2.5 volumes of ethanol, and 1 µL of GlycoBlueTM Coprecipitant (Ambion) overnight at −20 °C. After centrifugation at 4 °C for 1 h and washing with 70% ethanol, eccDNA was directly resuspended in the Illustra TempliPhi Sample Buffer and then amplified by random rolling circle amplification (RCA) using the Illustra TempliPhi Amplification Kit (GE Healthcare) according to the manufacturer's instructions, except that the reaction was incubated for 65 h at 28 °C. The enzyme was inactivated for 30 min at 70 °C. Out of 10 µL TempliPhi reaction, 1 µL was used to perform an EcoRI digestion (New England Biolabs) overnight at 37 °C. Quality control was performed by loading the EcoRI digestion product on a 1% agarose gel. The presence of a smear on the gel indicated a proper amplification step.

## Mobilome-seq with Illumina

For sequencing, 5 ng of amplified DNA (typically using a few µL of a 1/20 dilution of the TempliPhi amplification product) were used to prepare libraries using the Nextera XT kit, following the manufacturer's instructions. Twelve libraries were pooled per run. Sequencing was performed in house (UPVD Bio-Environment sequencing platform) using a Miseq with a 2 × 300 bp sequencing cassette. Quality control was performed on the resulting FASTQ files.

## Mobilome-seq with Nanopore

For ONT sequencing, 500 ng of amplified eccDNA from the TempliPhi product were used to prepare an ONT library using the Nanopore Rapid Sequencing Kit (SQK-RAD004) following the manufacturer's instructions. The libraries were sequenced in house (UPVD Bio-Environment sequencing platform) on ONT MinIONs using R9.4.1 flow cells. Basecalling was done on a GPU machine using Guppy with high-accuracy mode.

## Detection of eccDNAs from Illumina and ONT eccDNA-seq

The eccDNA producing loci from each *Arabidopsis* epigenetic mutant were detected using ecc_finder[6] with default parameters of short-read-mapping and long-read-mapping mode (for Illumina and ONT data, respectively). eccDNAs originating from organelle DNA fragments mapping to their nucleic copy (such as NUPTs for nucleoplasmic DNA and NUMTs for nuclear mitochondrial DNA), from ribosomal DNA repeats (rDNA) and from centromeric repeats were removed. The remaining eccDNAs were grouped into TE families by mapping them with BWA to the reference genome and normalized with FPKM (fragments per kilo base per million mapped reads, paired-end). For ONT eccDNA data, an additional counting of the unique reads was done. Reads from FASTQ files were mapped on the TAIR10 reference genome using minimap2 (version 2.24) with the following command $ mini-map2 map-ont -ax -t 8 -MD reference.fa file.fastq | samtools view -Sb -u |

samtools sort > $output.file.sort.bam. The bam file was converted into bed to collect the reads positions using bedtools (version 2.30.0). Unique reads were collected and the number of reads per TE or gene was calculated using bedtools intersect with the TE/gene annotation file and an homemade script. For each TE family the total number of unique reads was compiled using R for the 318 annotated TE families.

## EVD functional annotation

The two LTR sequences were identified by self-to-self alignment using BLAST. The sh-GAG structure of *EVD* was identified from long read RNA-seq[30]. RH sequence was identified using VLP data from the same *ddm1* mutant[50].

## Detection of truncated and chimeric eccDNAs from ONT eccDNA-seq

EccDNA-seq data were filtered for reads containing *EVD*. These *EVD* reads were mapped on the annotated structural domains of the *EVD* sequence using minimap2. Different profiles, such as the loss of different structural domains, were visualized using dot plot and then systematically grouped using bedtools groupby. In the next step, *EVD* reads were remapped to the reference genome using minimap2 to check for unmapped sequences. Chimeric eccDNAs supported by at least 5 reads were retained. The same approach was used for *ATCO-PIA21* eccDNAs.

## Genomic DNA sequencing with Nanopore

Genomic DNA was quality controlled using Nanodrop and Qubit quantifications. The libraries were sequenced in house (UPVD Bio-Environment sequencing platform) on ONT MinIONs using either R9.4.1 or R10 flow cells. Basecalling was done on a GPU machine using Guppy with high-accuracy mode.

## Detection of TE insertion polymorphisms from ONT genomic reads

Reads spanning the entire insertion and deletion sequence do not cause alignment breakpoints, but are flagged in the CIGAR. We developed a script to filter the CIGAR output. This pipeline can be found on GitHub (https://github.com/njaupan/CIGAR_SV). Briefly, in order to extract all reads containing insertions and deletions, we generated a PAF file from minimap2 (-cs -cx map-ont)[69]. The CIGAR metrics were then indexed until the position of the SV on the reference genome. Breakpoints with more than 4 supported reads at the same position were selected. The start and end positions of the breakpoints were extracted from the PAF file and grouped to generate common breakpoints displayed in BED format. The breakpoint locations were finally filtered for the presence of 5–20 bp TSDs supporting bona fide TE insertion polymorphisms.

## Identification of SVs

SVs from minimap2 alignments were detected using Sniffles v2.0[70], which generated a VCF file for each *A. thaliana* genotype separately. For duplications the VCF files were furthered filtered for duplications larger than 1 kb. Read depth was calculated with samtools depth and visualized with samplot. Reads spanning the junctions of the tandem duplications were extracted to manually validate the segmental duplications. For deletions the following parameter was used: -min-svlength 20 in order to recover small deletions for all samples. Deletions specific to the mutants were detected comparing the VCF files from the mutants versus the WT using bedtools intersect with -f =1E-9 (1 base overlap).

## Detection of DNA methylation from ONT genomic reads

Cytosine methylation patterns were detected from ONT reads using Nanopolish. The methylation patterns were parsed and plotted using methylartist (https://github.com/adamewing/methylartist).

### Re-analysis of DNAseq from met1-derived epiRILs

Raw data from DNAseq of *met1*-epiRILs was obtained from previously published work[41] (GEO ID: GSE120571) and re-analyzed. Briefly, after alignment with Bowtie2, peak calling was done for each epiRIL mapped bam file in comparison to the Col-0 control condition with MACS2 (https://github.com/taoliu/MACS/), setting the fragment size to 75 and extended size to 400 bp, with the options −B, -SPMR and −no model and Columbia-0 as control. The Log based background subtraction was performed with MACS2 bdgcmp with "logLR" as model and a *P*-value threshold of 0.00001.

### Plots and visualization

For the visualization samplot was used. For the scheme we used bioicons (https://bioicons.com/).

### Reporting summary

Further information on research design is available in the Nature Portfolio Reporting Summary linked to this article.

## Data availability

All high-throughput sequencing data generated in this study have been deposited at NCBI under Bioproject accession PRJNA956454. Source data are provided with this paper.

## Code availability

All custom scripts are available from Github [https://github.com/njaupan].

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

## Acknowledgements

We thank our EpiDiverse colleagues for stimulating discussions on eccDNAs, Drs. G. Moissiard, T. Lagrange and S. Jackson for fruitful comments on earlier versions of the manuscript and the platform "Bio-Environment" (UPVD, Perpignan, https://bio-environnement. univ-perp.fr/en) for sequencing. This work was supported by a grant from the French National Agency for Research (ANR-21-CE20-0047 CropCircle) and a European Training Network grant (EU H2020 Marie Skłodowska-Curie No 764965 EpiDiverse) to M.M.; a LABEX TULIP (ANR-10-LABX-41) and EUR TULIP-GS (ANR-18-EURE-0019) grants to M.M.'s host lab LGDP. M.C. is funded by the Royal Society Research Grant RGS\R1\201297 and UK Research through the Biotechnology and Biological Sciences Research Council (BBSRC) grant BB/W008866/1.

## Author contributions

P.Z. and M.M. designed the study. C.L. prepared plant material, extracted DNA and produced ONT data. P.Z., C.L., A.S., A.M. and M.M. performed eccDNA enrichment. P.Z., A.M., A.S. and M.M. performed computational analyses. R.K.S., A.G., M.I., F.P. and M.C. contributed material and data. P.Z. and M.M. prepared the manuscript. M.M., P.Z., F.P., R.K.S., and M.C. revised the manuscript. All authors read and approved the manuscript.

## Competing interests

The authors declare no competing interests.
