## [Peer Review File · Nature Communications]

Extrachromosomal circular DNA and structural variants highlight genome instability in Arabidopsis epigenetic mutantsReviewers' Comments:

Reviewer #1:

Remarks to the Author:

This paper presents an elegant analysis of active TEs in epigenetic mutants of Arabidopsis. Authors utilize combination of eccDNA sequencing together with long read Oxford Nanopore re-sequencing of individual mutant plants to show that only certain TEs are activated in DNA methylation mutants, show evidence of re-integration of circular TEs into the genome, and finally provide evidence for structural variation at the disease resistance locus by sequencing individual *ddm1* plants with long reads.

My major comment is about author's interpretation of their results. Current discussion states that 'transposon instability can lead to gene cluster duplication' inferring the cause and effect between TEs and duplication events. Although I agree that this is a parsimonious explanation, it is also possible that TEs preferentially insert in regions of instability while structural variation and duplications are caused by changes in DNA methylation regardless of TE insertion. In this case TEs serve as 'markers' of two speed genome, but are not the cause. I suggest to add a sentence to discussion to that regard.

Other major comments (that mostly need explanation):

- It is not clear why the authors did not include the *pollV rdr6* single mutants in their sequencing, especially as a control for their claims about the unique role of DDM1 as a epigenetic regulator of genomic stability. Also, the *ddm1 rdr6* double mutant showed fewer eccDNA TEs than the *ddm1* single mutant, and I would expect some intermediate between *ddm1* and the triple mutant. It would be useful to add what their hypothesis on this is.
- The long distribution of read lengths in Fig 1F is surprising. Why is there a bimodal peak for structure 3 and so many 10kb reads for structure 4? Is the grouping depicted in Fig 1E oversimplifying?
- In the results section, there is no comment on structural variation at RPP5 locus in the sequenced triple mutants. How pooling the 5 triple mutant plants impacted their assembly? The sibling *ddm1* plants had distinct insertion events. Was there any indication of an insertion present at 1/5th of the coverage, indicating it was unique to one plant? While being sympathetic to low input DNA, are you might be missing a fair amount of rearrangements by pooling. Did you try to look specifically at RPP5 locus to evaluate structural variation by mapping to reference and looking for split reads?
- In the discussion of genomic insertions, a dose dependent effect is suggested to explain why there are no insertions of COPIA and VANDAL despite their presence in eccDNAs. There are roughly twice as many EVD eccDNA reads relative to VANDAL. Is this reflected in the number insertions?

Minor comments:

- 'massive' in the title is a qualitative descriptor and can be omitted without loss of impact as it is not clear compared to what the identified events are massive. In many plants, structural variation has been recorded greater in scale or frequency.
- In methods, please add description of whole genome ONT sequencing, only eccDNAs. Please, add it, including coverage obtained and general assembly stats (Stable would be useful).

Reviewer #2:

Remarks to the Author:

The manuscript deals with the interplay of extrachromosomal circular DNA (eccDNA) and genome

stability. For this, four *Arabidopsis thaliana* mutants with relaxed control of transposition (*ddm1*; *ddm1 polIV*; *ddm1 rdr6*; *ddm1 polIV rdr6*) were analyzed. Their eccDNA levels were sequenced with Illumina for all mutants and with Oxford Nanopore for the triple mutant. This allowed understanding the eccDNA content, especially focusing on the presence of transposable elements (TEs) in full-length, with truncations, or re-organizations. To understand, how the TE-carrying eccDNAs relate to the mobilization and new integrations of TEs, re-sequencing of the mutants was performed. Then, the manuscript shifts to discussion of the re-sequenced mutants. New TE insertions are laid out (as expected), but then the manuscript drifts off to large-scale rearrangements without connecting it to the topic of the manuscript (eccDNAs). The manuscript finishes with a long discussion of the function of DDM1 and what can be learnt about DDM1 by re-sequencing the triple mutant. There is no going back to the experiment at hands or the eccDNAs.

Overall, the manuscript is very interesting and contains many new data and insights. However, the focus and presentation of data is confusing. The shift from eccDNAs to TEs to genomic rearrangements to *ddm1* functions is large and there is no circling back to the eccDNAs. Hence, beginning and end of the manuscript do not match and the introduction only focuses on part 1 (eccDNA, transposable element mobilization). How do the sequenced eccDNAs relate to the large genome duplications? This is for me a major point, as this leaves the manuscript very unsharp. I think that it is not publishable in the current state.

Nevertheless, this manuscript contains many new, interesting, and certainly publishable data. I could not identify any flaws in the methods. Sequence data has been made available. The references are comprehensive.

Point-by-point discussion:

Major point:

1) Structure and framing of the manuscript is not focused. Begin and end do not match, as outlined above. It is not clear what the manuscript wants; it is more a step-by-step recounting of the performed experiments, their outcome, and the shifts in interests (begin  eccDNAs  TEs  resequencing  TE integrations  genome rearrangements  DDM1 functions  end), but it does not have a well-defined message.

Minor points/clarifications:

2) How does the Illumina analysis compare with the ONT sequencing of the eccDNA circles?

3) Line 105: Are the mentioned eccDNAs of EVD, VANDAL21, ATCOPIA51, ATCOPIA52 and VANDAL3 the only eccDNAs sequenced?

4) Figure 1: Have there also been circles sequenced with only a single LTR as has been previously proposed in the literature?

5) Line 127: The text now mentions ATCOPIA21, but it did not appear in the listing before. Should it be added? It is a bit confusing to understand where this comes from.

6) Line 138: Did the truncated de novo TE insertions correspond to the truncated eccDNAs?

7) Lines 140 and following: The scenarios are not quite clear and especially how they link to the formation of eccDNAs. Also, they may be more fitting in the discussion section, as they do not influence the next experiments.

8) As the mentioned scenarios above, in many instances, discussion points are added to the results, e.g. in line 129, line 177, line 184; shifting those to the discussion would make it easier to grasp the

structure/message.

9) Line 163: The description of the "3-hit" reads is unclear. The corresponding figures 3b-d were also not clear to me.

10) Line 169: A 9-bp TSD is very untypical for an LTR retrotransposon (usually they span 5 bp). Is this normal for ATCOPIA21 or how could this be explained?

11) For re-sequencing of the triple-mutant, 5-6 pooled plants were used. How are these plants related and does this setup provide limitations, e.g. for the identification of new TE integration events?

12) Discussion: Overall, the fact that the insertion of many truncated TEs was observed is certainly interesting. However, this is not new, so I suggest to tone down the novelty of this fact a bit and to also cite some of the papers that have described truncation and TE mechanisms. There are many important insights taken from the paper, but the truncation may be the least surprising. If the manuscript highlights preferred breaking points or typical truncation structures, this would be really interesting for many (and I think that the data for this is at hand). I also think that the fusions and chimeric TEs could be highlighted more.

13) Discussion: absence of eccDNAs in the discussion after the first lines, despite this being the main focus of the manuscript. (This is my main point as I have also indicated above)

Figures/Legends:

14) Figure 1F: I have the feeling that I am missing something: How is the read length relevant for the structure of the circle? Is the shown distribution instead the length of the circle?

15) All figures: The colors in the figures are in most cases not explained in the legend.

16) Figure 3d: I could not identify the typical TG/CA delimiting an LTR. Maybe this can be added?

17) Figure 3b: "I" and "N" are unclear.

18) Figure 6 is very helpful and the manuscript would profit for a stronger focus on this. It has the potential to link together the manuscript.

Summarizing, the manuscript presents many interesting data, but suffers at stringing them together. In my eyes, this goes beyond a revision. I recommend rejection with resubmission invited.

Reviewer #3:

Remarks to the Author:

This manuscript describes several analyses of eccDNA and structural variation in Arabidopsis plants defective in components of DNA methylation machinery. There are some very interesting observations in this study and I was quite interested in the topic. However, I found the results very difficult to comprehend based on the current descriptions and visualizations. I found it difficult to fully evaluate the results and potential impact of this work given the lack of key details.

Missing information:

1. Biological materials. Prior studies have found evidence for changes in ddm1 mutants after several generations of propagating these materials in a homozygous condition. In this study it is not clear whether the materials are first generation homozygotes or later generation materials. In addition, the long read analyses focused on two ddm1 plants (separate sequencing) and two pools of ddm1, PolIV, rdr6 plants. What is the relationship of these plants to each other? Are they all siblings? What was the

parental genotype?

2. Sequencing details. I could not find information on the depth of sequencing or read length distribution for either the eccDNA or the genomic samples. It is important to provide this information to assess the likely power to detect structural variants. In addition, it was not clear if there was any biological replication of the eccDNA sequencing. I think the conclusions are based on a single replicate for each genotype but this is not clear.

3. Figure details. Many of the figures in this work are screenshots of IGV with somewhat limited description and resolution. I will provide specific comments below but in many cases these figures were not sufficient support of the statements made in the results section.

Specific comments:

Lines 95-97 (Figure 1A): Do the authors have any explanation for why there is substantially less eccDNA for the highlighted elements in *ddm1* polIV double mutants relative to *ddm1* mutants? It was really not clear that the double or triple mutants were having significant effects compared to the *ddm1* mutant. I doubt the authors could make any definitive statements or interpretations on this due to the lack of biological replication for the experiment. However, it seems plausible that only *ddm1* matters for the observed phenotypes.

Line 110: In figure 1E the structure 1 seems to be an eccDNA of the full EVD element. Can the authors clarify that this has two full length LTRs. I think that is what is being shown but it wasn't clear if there was one or two copies of the LTR sequence in this structure. Also, in the dotplot alignments it seems that most of the reads have multiple copies of the sequence. I inferred that this is due to multiple reads around the circular structure. However, it was unclear if that was the case. It could be useful to provide some description of the eccDNA / mobilome sequencing protocol to clarify the expectations in terms of whether you expect a single pass over a circular molecule or multiple passes around the circular structure.

Line 111-119: I struggled to understand the data presented in figure 1D and F together. For example, structure 1 has the full EVD structure. How can you have a read that support this structure that is only ~2,000bp in length? The read distribution has many various read lengths (shorter or longer) and I did not understand how you could confidently assign short reads to one specific structure.

Line 122: The authors seek to detect new TE insertions using long read sequencing. What coverage was obtained and what was the read length distribution. What is the zygosity for these insertions? How many are heterozygous? In particular, for the triple mutant the authors are sequencing a pool of multiple plants – how common would the insertion need to be in order to be detected? Could one of six plants be hemizygous and you would be able to detect the insertion?

Line 124: The authors should specify in the text and figure 2A that the comparison of numbers is perhaps not informative in terms of comparing movement rates since the *ddm1* and double mutants are single plants while the triple mutant is a pool of plants. Also, I could not tell if figure 2A was based on a single sequencing or sequencing of multiple plants as described later in the section on duplications?

Line 138-140. The authors claim that Figure 1E provides evidence of TSDs. I don't see how I can see this evidence of a target site duplication based on the presented information.

Line 148-149: Two comments about this section. First, the authors highlight that there are 7 copies in a single long read. Does this truly reflect 7 copies or is this just the result of 7 passes around a circular molecule? Second, how are the authors using the term chimeric here. I initially thought they were referring to molecules resulting from fusion of distinct DNA pieces within eccDNA. However, in this case it seems that this is the result of formation of an eccDNA that contains a portion of EVD

along with some flanking sequence.

Figure 3D – How does this visual provide evidence for a TSD? Why is one read highlighted in orange. Perhaps this is showing the same read has both copies of the TSD? That is not clear though in the visual since I could not tell if these are the same reads with two alignments or distinct reads that overlap the region.

Lines 169-171: The authors assert that a structural rearrangement results in a duplication of RPP5 with a new copy closer to CRK19. Is there actual evidence for this? Figure 4D is a schematic but this is labeled as a model. I could not find any evidence that actually showed that there is a true RPP5 duplication / movement event.

Lines 173-186: I got a little confused in this section. The authors look at a set of 64 lines and find many examples of EVD or ONSEN insertions. However, there was not a control. If the authors look at another region of similar size and gene density how many of the accessions would have an EVD or ONSEN insertion? It is quite possible the observed results represent an increased rate of insertions but that is not currently demonstrated. Second, the authors then look at 21 epiRILs and find evidence that nearly half have lost RPP5. This is kind of the opposite of what the authors found – duplications at this locus. This seems to suggest common loss of RPP5 in epiRILs but frequent duplications of the locus in ddm1 mutants – I think but I might be missing something here.

Lines 189-191: The authors perform de novo genome assembly of the ddm1 and triple mutant. I think they are doing this separately for each sequenced sample (individual plants for ddm1 and pooled samples for triple mutant). They mention that they found two duplications in ddm1. Were both of these detected in both of the siblings? Are these homozygous duplications or heterozygous? I could not tell from the methods section how heterozygosity was handled in the assembly. This becomes especially important in the triple mutant which may have segregation for novel structural variants. It wasn't clear but it seems that no structural variants were detected in the triple mutant, even though there was deeper sequencing of this genotype.

Lines 202-204: The authors claim a difference in DNA methylation but this really was not clear from the figure. The potential changes in methylation were fairly subtle and not convincing.

REVIEWER COMMENTS

#Reviewer #1:

This paper presents an elegant analysis of active TEs in epigenetic mutants of Arabidopsis. Authors utilize combination of eccDNA sequencing together with long read Oxford Nanopore re-sequencing of individual mutant plants to show that only certain TEs are activated in DNA methylation mutants, show evidence of re-integration of circular TEs into the genome, and finally provide evidence for structural variation at the disease resistance locus by sequencing individual *ddm1* plants with long reads.

My major comment is about #author's interpretation of their results.

Current discussion states that 'transposon instability can lead to gene cluster duplication' inferring the cause and effect between TEs and duplication events. Although I agree that this is a parsimonious explanation, it is also possible that TEs preferentially insert in regions of instability while structural variation and duplications are caused by changes in DNA methylation regardless of TE insertion.

In this case TEs serve as 'markers' of two speed genome, but are not the cause.

I suggest to add a sentence to discussion to that regard.

We thank the reviewer for pointing out this cause/effect issue, we agree with this and have modified the text accordingly.

Other major comments (that mostly need explanation):

#1• It is not clear why the authors did not include the *pollV rdr6* single mutants in their sequencing, especially as a control for their claims about the unique role of DDM1 as a epigenetic regulator of genomic stability. Also, the *ddm1 rdr6* double mutant showed fewer eccDNA TEs than the *ddm1* single mutant, and I would expect some intermediate between *ddm1* and the triple mutant. It would be useful to add what their hypothesis on this is.

Indeed, we agree with the reviewer that single mutants could add some information. We have added the *pollV* and *rdr6* single mutants as controls in the mobilome-seq experiment. Additionally we have added some explanation on why the *ddm1 rdr6* show fewer eccDNA TEs together with fewer integrations, citing Tsukahara et al., (2009) who reported that different lines of *ddm1* had different relative TE activities.

#2• The long distribution of read lengths in Fig 1F is surprising. Why is there a bimodal peak for structure 3 and so many 10kb reads for structure 4? Is the grouping depicted in Fig 1E oversimplifying?

We realized that this figure was confusing and depended on the technical characteristics of the sequencing run. We have thus modified the Figure. Please note that for clarity (and to answer another reviews comments) we have now separated the full length ecccDNA and insertions (Fig. 1) from the truncated (Fig. 2) and chimeric ones (Fig. 3).

#3• In the results section, there is no comment on structural variation at *RPP5* locus in the sequenced triple mutants. How pooling the 5 triple mutant plants impacted their assembly? The sibling *ddm1* plants had distinct insertion events. Was there any indication of an insertion present at 1/5th of the coverage, indicating it was unique to one plant? While being sympathetic to low input DNA, you might be missing a fair amount of rearrangements by pooling. Did you try to look specifically at *RPP5* locus to evaluate structural variation by mapping to reference and looking for split reads?

It is indeed a very good point. Following the reviewer's advice, we have now looked at the *RPP5* locus in the triple mutant (ONT reads). In one out of 2 biological replicates we have found a 13.3kb deletion, supported by 3 reads. Surprisingly the deletion is located at the same position than in the *met1* derived epiRILs. This confirms that this locus is prone to SVs and that deletion is most likely mediated by small homologies in this highly repeated locus (see dotter in Supplementary Fig. 7). We double checked the pooling of the samples and actually 3 plants (not 5) were pooled for the triple mutant. We apologize for this mistake. At the *RPP5* locus, the fact that 3 reads only support the deletion (average coverage of 50X) indicates that out of our pool of 3 plants, only one might be heterozygous for the deletion, or that the deletion could be somatic.

#4• In the discussion of genomic insertions, a dose dependent effect is suggested to explain why there are no insertions of *COPIA* and *VANDAL* despite their presence in eccDNAs. There are roughly twice as many *EVD* eccDNA reads relative to *VANDAL*. Is this reflected in the number insertions?

In the triple mutant there is roughly a ten fold difference in terms of new insertions from 73 insertions for *EVD* to 6 insertions for *VANDAL21* (now Fig1). In terms of eccDNA, the amount of *EVD* reads can vary between two replicates (see new Supplementary Fig. 3), the activity of *EVD* has also been previously shown to vary between *ddm1* plants (Tsukuhara et al., Nature 2009). In the ONT mobilome-seq data we observed a 1.2 to 13 fold difference between *EVD* and *VANDAL21* reads (Supplementary Fig. 3). Nonetheless, we found insertions of both *EVD* and *VANDAL21*.

Concerning *COPIA21* no eccDNA could be detected despite the presence of new insertions. *COPIA21* could make eccDNA in a tissue specific manner, or with a very limited short life, for yet unknown reasons, limiting our power to characterize them.

Minor comments:

#5• 'massive' in the title is a qualitative descriptor and can be omitted without loss of impact as it is not clear compared to what the identified events are massive. In many plants, structural variation has been recorded greater in scale or frequency.

We agree and have deleted « massive » from the the title (new title: Genome instability in Arabidopsis epigenetic mutants revealed by extrachromosomal circular DNA and structural variants detection). This term was coined when we initially were intrigued by the rearrangements non previously identified in the genome of the triple mutant.

#6• In methods, please add description of whole genome ONT sequencing, only eccDNAs. Please, add it, including coverage obtained and general assembly stats (Stable would be useful).

Thank you for the suggestion. We apologize for this, we have added a full description of the methods and now provide all runs characteristics as Supplementary Tables.

Reviewer #2 (Remarks to the Author):

The manuscript deals with the interplay of extrachromosomal circular DNA (eccDNA) and genome stability. For this, four *Arabidopsis thaliana* mutants with relaxed control of transposition (*ddm1*; *ddm1 polIV*; *ddm1 rdr6*; *ddm1 polIV rdr6*) were analyzed. Their eccDNA levels were sequenced with Illumina for all mutants and with Oxford Nanopore for the triple mutant. This allowed understanding the eccDNA content, especially focusing on the presence of transposable elements (TEs) in full-length, with truncations, or re-organizations. To understand, how the TE-carrying eccDNAs relate to the mobilization and new integrations of TEs, re-sequencing of the mutants was performed. Then, the manuscript shifts to discussion of the re-sequenced mutants. New TE insertions are laid out (as expected), but then the manuscript drifts off to large-scale rearrangements without connecting it to the topic of the manuscript (eccDNAs). The manuscript finishes with a long discussion of the function of DDM1 and what can be learnt about DDM1 by re-sequencing the triple mutant. There is no going back to the experiment at hands or the eccDNAs.

Overall, the manuscript is very interesting and contains many new data and insights. However, the focus and presentation of data is confusing. The shift from eccDNAs to TEs to genomic rearrangements to *ddm1* functions is large and there is no circling back to the eccDNAs. Hence, beginning and end of the manuscript do not match and the introduction only focuses on part 1 (eccDNA, transposable element mobilization). How do the sequenced eccDNAs relate to the large genome duplications? This is for me a major point, as this leaves the manuscript very unsharp. I think that it is not publishable in the current state.

We would like to thank the reviewer for his/her suggestion to reshape the manuscript. The initial writing took into account our initial surprise after serendipitously characterizing so many chimeric reads in the genome of the triple mutant. For clarity we have now reformatted completely the text and figures of the manuscript, focusing on eccDNAs. We hope that the current version will be more straightforward. Additionally we have added a new hypothesis linking eccDNA and SVs. We suggest that the large amount of DSBs present in *ddm1* and its mutant combinations could

overload the DNA repair machinery, triggering less common repair pathways such as the pol theta pathway recently described in mammals and that lead to SVs, notably tandem duplications. We leave the discussion open but believe it is a field to be explored further.

Nevertheless, this manuscript contains many new, interesting, and certainly publishable data. I could not identify any flaws in the methods. Sequence data has been made available. The references are comprehensive.

Point-by-point discussion:

Major point:

#1) Structure and framing of the manuscript is not focused. Begin and end do not match, as outlined above. It is not clear what the manuscript wants; it is more a step-by-step recounting of the performed experiments, their outcome, and the shifts in interests (begin  eccDNAs  TEs  resequencing  TE integrations  genome rearrangements  DDM1 functions  end), but it does not have a well-defined message.

We agree with the reviewer's comment. We have clarified our message and summarize the new manuscript as follows: begin: eccDNAs and genome interaction  resequencing eccDNA+TE integrations  finding truncated eccDNA/TE integrations  finding chimeric eccDNA/TEs  finding genome rearrangements in high eccDNA lines  discussing possible impact of eccDNA on DNA repair end'. We hope that the new version will have helped clarify the message and address the reviewer's comment.

Minor points/clarifications:

#2) How does the Illumina analysis compare with the ONT sequencing of the eccDNA circles?

The ONT sequencing clearly outperforms the Illumina. First the coverage at each eccDNA producing locus is less biased (see for instance Fig. 1C at *EVD* locus). More importantly the length of the reads allows to reconstruct the entirety of the circles, defining gaps (see new Fig. 2B) and chimeric sequences (new Fig. 3A). Because Illumina is still cheaper, for a new eccDNA study we would recommend a first screen with Illumina (to select the samples abundant in circles) and a second one on the samples with highest eccDNA load with ONT, in order to better describe the circles.

#3) Line 105: Are the mentioned eccDNAs of *EVD*, *VANDAL21*, *ATCOPIA51*, *ATCOPIA52* and *VANDAL3* the only eccDNAs sequenced?

We provide a list of all TE-eccDNAs sequenced (Supplementary Table 3), indicating the number of covered loci for each sample and the number of unique reads covering these loci, for all TE

families. The ones described in the Figures and text are the most represented and the most enriched compared to WT. Of note *ATCOPIA51* and *ATCOPIA52* are not enriched in the second biological replicate. *ATREP10D* and *HELITRONY3* are also covered by reads but these families are represented by hundreds of loci in the genome and the number of reads is not enriched in the triple mutant compared to the WT.

#4) Figure 1: Have there also been circles sequenced with only a single LTR as has been previously proposed in the literature?

Yes, indeed we found this type of circle (also very frequent in retroviruses derived eccDNAs). We have now indicated this information in Fig 2.

#5) Line 127: The text now mentions *ATCOPIA21*, but it did not appear in the listing before. Should it be added? It is a bit confusing to understand where this comes from.

We apologize for the confusion. *ATCOPIA21* is indeed present as new insertions but we could not detect it as eccDNA. We hypothesize that *COPIA21* could make eccDNA in a tissue specific manner, limiting our power to characterize them. For instance, concerning class II elements, *Mu* and *Ds* elements preferentially insert in genes expressed in meristematic enriched tissues (Zhang et al, NAR 2020, <https://doi.org/10.1093/nar/gkaa370>).

#6) Line 138: Did the truncated de novo TE insertions correspond to the truncated eccDNAs?

No, we found evidence of both truncated eccDNA and neo-insertions but for instance neo-insertions always include 2 LTRs whereas eccDNA can miss one LTR. We have now clarified this point (LINE).

#7) Lines 140 and following: The scenarios are not quite clear and especially how they link to the formation of eccDNAs. Also, they may be more fitting in the discussion section, as they do not influence the next experiments.

We have now reformulated this and moved to the discussion part.

#8) As the mentioned scenarios above, in many instances, discussion points are added to the results, e.g. in line 129, line 177, line 184; shifting those to the discussion would make it easier to grasp the structure/message.

We have now moved all parts to the discussion, thank you for the suggestion.

#9) Line 163: The description of the “3-hit” reads is unclear. The corresponding figures 3b-d were also not clear to me.

We have clarified Fig. 3. Fig. 3B now contains a scheme describing the pipeline to look for eccDNAs containing chimeric *EVD* (meaning *EVD* plus another unrelated sequence). In 3B we show the IGV view of the *AT5G66440* locus with the proper WT control (ONT reads). In 3C we show the location of the insertions of *EVD* in this genome (term « genome » added) and in 3D we

highlight the position of the TSD corresponding to the *EVD* insertion at the *AT5G66440* locus. The figure legend was edited accordingly.

#10) Line 169: A 9-bp TSD is very untypical for an LTR retrotransposon (usually they span 5 bp). Is this normal for *ATCOPIA21* or how could this be explained?

We agree this is a bit unusual. We did not find previous reports for specific *ATCOPIA21* TSDs. A recent report suggest that TSDs can be 3-10bp long (Miyao, A., Yamanouchi, U. Transposable element finder (TEF): finding active transposable elements from next generation sequencing data. *BMC Bioinformatics* **23**, 500 (2022). <https://doi.org/10.1186/s12859-022-05011-3>). We have added this reference in the text.

We had a closer look at the reference genome (TAIR10), where there is only one full length copy of *ATCOPIA21* (*AT5TE65370*), with 120bp long LTRs (100% identity). These LTRs start and end with TG-TA (instead of TG-CA). The TSD at this locus is 5bp long (AAATC) but because the element has inserted into a sequence starting with TA and ending with T the apparent TSD is TAAAATCT (8bp). Such a coincidence could also explain the 9bp apparent TSD (TATAGTAGC) at the new insertion locus (Fig. 4C).

#11) For re-sequencing of the triple-mutant, 5-6 pooled plants were used. How are these plants related and does this setup provide limitations, e.g. fo the identification of new TE integration events?

These plants are siblings, we added this description in the Methods. This set up provides limitations in our resolution to identify new TE insertions. Our designs allows to identify the TE insertions inherited from the mother plant, we make this limitations clear in the text. However given the very dwarf phenotype of the plants we could not easily down scale the DNA extraction for a proper ONT sequencing run. In the near future the advance of the technology will probably allow us to identify single plant insertions even for this triple mutant.

#12) Discussion: Overall, the that the insertion of many truncated TEs was observed is certainly interesting. However, this is not new, so I suggest to tone down the novelty of this fact a bit and to also cite some of the papers that have described truncation and TE mechanistics. There are many important insights taken from the paper, but the truncation may be the least surprising. If the manuscript highlights preferred breaking points or typical truncation structures, this would be really interesting for many (and I think that the data for this is at hands). I also think that the fusions and chimeric TEs could be highlighted more.

We have toned down the claim that truncated insertions are novel. We are aware of the important datasets created when using TEs to create mutation lines implants (e.g. *Tos17*), however the fact that truncation is already observed at the eccDNA level seems interesting. We also have better described the chimeras.

We added the following references showing truncated TE insertions and rearrangements due to TEs:

- Vitte C, Panaud O, Quesneville H. LTR retrotransposons in rice (*Oryza sativa*, L.): recent burst amplifications followed by rapid DNA loss. *BMC Genomics*. 2007 Jul 6;8:218. doi: 10.1186/1471-2164-8-218.
- Chaparro C, Guyot R, Zuccolo A, Piégu B, Panaud O. RetrOryza: a database of the rice LTR-retrotransposons. *Nucleic Acids Res*. 2007 Jan;35(Database issue):D66-70. doi: 10.1093/nar/gkl780.
- Xuan YH, Piao HL, Je BI, Park SJ, Park SH, Huang J, Zhang JB, Peterson T, Han CD. Transposon Ac/Ds-induced chromosomal rearrangements at the rice OsRLG5 locus. *Nucleic Acids Res*. 2011 Dec;39(22):e149. doi: 10.1093/nar/gkr718.
- Wang D, Zhang J, Zuo T, Zhao M, Lisch D, Peterson T. Small RNA-Mediated De Novo Silencing of Ac/Ds Transposons Is Initiated by Alternative Transposition in Maize. *Genetics*. 2020 Jun;215(2):393-406. doi: 10.1534/genetics.120.303264.

#13) Discussion: absence of eccDNAs in the discussion after the first lines, despite this being the main focus of the manuscript. (This is my main point as I have also indicated above)

We have focused the discussion around the eccDNA as suggested.

Figures/Legends:

#14) Figure 1F: I have the feeling that I am missing something: How is the read length relevant for the structure of the circle? Is the shown distribution instead the length of the circle?

This point was also raised by another reviewer, we realized that this figure panel was not relevant and replaced it with a clear picture of the deleted structures.

#15) All figures: The colors in the figures are in most cases not explained in the legend.

Thank you, we have edited the legends to include the color description.

#16) Figure 3d: I could not identify the typical TG/CA delimitating an LTR. Maybe this can be added?

Sorry if the figure was confusing, what is shown is the TSD (target site duplication) and thus does not contain the LTR sequence, that's why TG/CA cannot be found. We have clearly outlined the TSD to avoid this confusion.

#17) Figure 3b: "I" and "N" are unclear.

We have simplified this and only ONT reads (previously « N » for Nanopore) are shown.

18) Figure 6 is very helpful and the manuscript would profit for a stronger focus on this. It has the potential to link together the manuscript.

Thank you for pointing this out. We have added a better description of it in the text. We have also included our hypothesis on the link between a high eccDNA load associated with high TE activity and high DSB number with the impact on DNA repair and SVs.

Summarizing, the manuscript presents many interesting data, but suffers at stringing them together. In my eyes, this goes beyond a revision. I recommend rejection with resubmission invited.

Reviewer #3 (Remarks to the Author):

This manuscript describes several analyses of eccDNA and structural variation in Arabidopsis plants defective in components of DNA methylation machinery. There are some very interesting observations in this study and I was quite interested in the topic. However, I found the results very difficult to comprehend based on the current descriptions and visualizations. I found it difficult to fully evaluate the results and potential impact of this work given the lack of key details.

We would like to thank the reviewer for his/her interest and have tried to give more details in the revised version.

Missing information:

#1. Biological materials. Prior studies have found evidence for changes in *ddm1* mutants after several generations of propagating these materials in a homozygous condition. In this study it is not clear whether the materials are first generation homozygotes or later generation materials. In addition, the long read analyses focused on two *ddm1* plants (separate sequencing) and two pools of *ddm1*, *PollIV*, *rdr6* plants. What is the relationship of these plants to each other? Are they all siblings? What was the parental genotype?

We used first generation *ddm1* and the plants used are siblings. This is now specified in the material and methods section.

#2. Sequencing details. I could not find information on the depth of sequencing or read length distribution for either the eccDNA or the genomic samples. It is important to provide this information to assess the likely power to detect structural variants. In addition, it was not clear if there was any biological replication of the eccDNA sequencing. I think the conclusions are based on a single replicate for each genotype but this is not clear.

We have added replicates for the eccDNA sequencing. We have added two tables summarizing the technical characteristics of each sequencing run including the coverage (Supplementary Tables 1 and 2 for Illumina and ONT, respectively).

#3. Figure details. Many of the figures in this work are screenshots of IGV with somewhat limited description and resolution. I will provide specific comments below but in many cases these figures were not sufficient support of the statements made in the results section.

We agree that IGV screenshots are not ideal but they allow to visualize the reads. We have added more samples with samplots (Fig. 1C, Supplementary Fig. 2B).

Specific comments:

#4. Lines 95-97 (Figure 1A): Do the authors have any explanation for why there is substantially less eccDNA for the highlighted elements in *ddm1 polIV* double mutants relative to *ddm1* mutants? It was really not clear that the double or triple mutants were having significant effects compared to the *ddm1* mutant. I doubt the authors could make any definitive statements or interpretations on this due to the lack of biological replication for the experiment. However, it seems plausible that only *ddm1* matters for the observed phenotypes.

We have performed additional replicates for both the eccDNA sequencing and the genome sequencing (summarized in Supplementary Tables 1 and 2). Our conclusion is that the *ddm1 rdr6* double mutant has reproducibly less eccDNA and insertions than *ddm1* single or *ddm1* in combination with *polIV*. The reason behind this phenotype is currently unknown.

#5. Line 110: In figure 1E the structure 1 seems to be an eccDNA of the full EVD element. Can the authors clarify that this has two full length LTRs. I think that is what is being shown but it wasn't clear if there was one or two copies of the LTR sequence in this structure. Also, in the dotplot alignments it seems that most of the reads have multiple copies of the sequence. I inferred that this is due to multiple reads around the circular structure. However, it was unclear if that was the case. It could be useful to provide some description of the eccDNA / mobilome sequencing protocol to clarify the expectations in terms of whether you expect a single pass over a circular molecule or multiple passes around the circular structure.

We thank the reviewer for pointing this out. To clearly illustrate the protocols that we used we have added a Supplementary Fig 1. Indeed because of the Phi29, the circles are amplified and several repeats of the circle can be found in a unique ONT read. Concerning the 1 or 2 LTR structure of the circle, we have now clarified this in Fig. 2B. Half of the circles are either 1LTR or 2LTRs ones, the rest being partial copies.

#6. Line 111-119: I struggled to understand the data presented in figure 1D and F together. For example, structure 1 has the full EVD structure. How can you have a read that support this structure that is only ~2,000bp in length? The read distribution has many various read lengths (shorter or longer) and I did not understand how you could confidently assign short reads to one specific structure.

We apologize as we realized that Fig. 1F was not relevant to describe the structure of the circles (also pointed out by other reviewers). We have now clarified this in the new Fig. 2 showing the truncated eccDNA together with the truncated insertions.

#7. Line 122: The authors seek to detect new TE insertions using long read sequencing. What coverage was obtained and what was the read length distribution. What is the zygosity for these insertions? How many are heterozygous? In particular, for the triple mutant the authors are sequencing a pool of multiple plants – how common would the insertion need to be in order to be detected? Could one of six plants be hemizygous and you would be able to detect the insertion?

We have added the full description of the sequencing runs as Supplementary Table 1 and 2. Out of our 13 resequencing samples we have a coverage ranging from 20 to 99 fold. Most insertions are

heterozygous, we have added the information relative to number of reads in Supplementary Table 4. The fact that we detect a deletion at *RPP5* in one replicate of the triple mutant genome (pool of 3 plants) suggest that we can detect SVs in a pool, with sufficient coverage (here 50X).

#8. Line 124: The authors should specify in the text and figure 2A that the comparison of numbers is perhaps not informative in terms of comparing movement rates since the *ddm1* and double mutants are single plants while the triple mutant is a pool of plants. Also, I could not tell if figure 2A was based on a single sequencing or sequencing of multiple plants as described later in the section on duplications?

It is a good point, we have edited the Figure legend and text accordingly to clarify the number of plants that was sequenced (one plant except for the triple mutant where we sequenced a pool of 3 siblings). Because an heterozygous insertion in one of the pooled siblings would lead to a SV supported by only a few reads, the list of TE insertions for this sample reflect the insertions in the mother plant. We applied a threshold of 4 reads for the insertions, we now provide the number of supporting reads and total reads spanning each new insertion as Supplementary Table 5.

#9. Line 138-140. The authors claim that Figure 1E provides evidence of TSDs. I don't see how I can see this evidence of a target site duplication based on the presented information.

We apologize for the confusion, the former Fig 1E did not show TSD, instead Fig. 3D showed a TSD (see #11 below).

#10. Line 148-149: Two comments about this section. First, the authors highlight that there are 7 copies in a single long read. Does this truly reflect 7 copies or is this just the result of 7 passes around a circular molecule? Second, how are the authors using the term chimeric here. I initially thought they were referring to molecules resulting from fusion of distinct DNA pieces within eccDNA. However, in this case it seems that this is the result of formation of an eccDNA that contains a portion of EVD along with some flanking sequence.

As stated above the 7 copies correspond to 7 passes around the circular molecule. The term chimeric is used here to describe a chimeric circularization forming eccDNA containing sequences corresponding to 2 different locations in the genome (for a review see Yang L, Jia R, Ge T, Ge S, Zhuang A, Chai P, Fan X. Extrachromosomal circular DNA: biogenesis, structure, functions and diseases. *Signal Transduct Target Ther.* 2022 Oct 2;7(1):342. doi: 10.1038/s41392-022-01176-8).

#11. Figure 3D – How does this visual provide evidence for a TSD? Why is one read highlighted in orange. Perhaps this is showing the same read has both copies of the TSD? That is not clear though in the visual since I could not tell if these are the same reads with two alignments or distinct reads that overlap the region.

The ONT reads shown in this figure all start or end at the same position. This is the case when you have an insertion not present in the reference genome (hence only the part of the read properly mapping to the genome is shown, the rest of the read being omitted). Furthermore the coverage is higher at the TSD because there is actually twice this sequence in the sequenced genome versus

once in the reference. We have now clearly indicated the position of the TSD on the figure and corresponding legend.

#12. Lines 169-171: The authors assert that a structural rearrangement results in a duplication of RPP5 with a new copy closer to CRK19. Is there actual evidence for this? Figure 4D is a schematic but this is labeled as a model. I could not find any evidence that actually showed that there is a true RPP5 duplication / movement event.

Because we have focused the revised manuscript on eccDNA we now provide this Fig. as Supplementary Fig. 6. We have added a biological replicate for the triple mutant and could confirm this chimerism at the *RPP5* locus (mentioned in Supplementary Fig. 6). The legend has been changed to « possible scenario ». Indeed while the rearrangement was detected, its precise origin can only be inferred. We still think that this observation could be of interest to our colleagues interested in genome dynamics at the *RPP5* locus.

#13. Lines 173-186: I got a little confused in this section. The authors look at a set of 64 lines and find many examples of EVD or ONSEN insertions. However, there was not a control. If the authors look at another region of similar size and gene density how many of the accessions would have an EVD or ONSEN insertion? It is quite possible the observed results represent an increased rate of insertions but that is not currently demonstrated. Second, the authors then look at 21 epiRILs and find evidence that nearly half have lost RPP5. This is kind of the opposite of what the authors found – duplications at this locus. This seems to suggest common loss of RPP5 in epiRILs but frequent duplications of the locus in *ddm1* mutants – I think but I might be missing something here.

For clarity we have decided to delete this analysis concerning the accumulation of *EVD* and *ONS* at the *RPP5* locus. Instead (and following another reviewer's suggestion) we looked at *RPP5* stability in our own dataset of genomic sequences. We found evidence that the *RPP5* is deleted in at least one plant of the pooled triple mutant sample (see Supplementary Fig. 6). The reviewer is right that unstable loci can display either duplications or gene loss. While the mechanisms must be different, what we wanted to highlight here is the general instability of some genes. We have toned down the importance of this observation but we still think this could be of interest to our colleagues interested in genome dynamics at pathogen response genes.

#14. Lines 189-191: The authors perform de novo genome assembly of the *ddm1* and triple mutant. I think they are doing this separately for each sequenced sample (individual plants for *ddm1* and pooled samples for triple mutant). They mention that they found two duplications in *ddm1*. Were both of these detected in both of the siblings? Are these homozygous duplications or heterozygous? I could not tell from the methods section how heterozygosity was handled in the assembly. This becomes especially important in the triple mutant which may have segregation for novel structural variants. It wasn't clear but it seems that no structural variants were detected in the triple mutant, even though there was deeper sequencing of this genotype.

Genome assembly was initially used to detect SVs. Thanks to the development of more powerful tool such as Sniffles2 during the revisions, we were able to accurately detect SVs without the need

for assembly. We have clarified this in the methods. We found large tandem duplications in *ddm1* and *ddm1 polIV* but not in the triple mutant. We hypothesize that pooling the plants in the triple mutant has decreased our power to detect SVs.

#15. Lines 202-204: The authors claim a difference in DNA methylation but this really was not clear from the figure. The potential changes in methylation were fairly subtle and not convincing.

We agree and did not mean to make a big case of this difference, we have moved this Figure to Supplementary Fig. 9, just for information.

Reviewers' Comments:

Reviewer #1:

Remarks to the Author:

Manuscript has been improved in clarity of data presentation, the authors addressed all my comments. I am very happy to see this work!

In my opinion, the authors also addressed comments from the second reviewer including focusing the article, providing additional supplemental data, clarifying the figures and adjusting the discussion.

Reviewer #3:

Remarks to the Author:

The authors revisions have clarified some of the key datasets and results and have improved the presentation of the results. These revisions have satisfied that concerns that I had on the first draft of the manuscript.

REVIEWERS' COMMENTS

Reviewer #1 (Remarks to the Author):

Manuscript has been improved in clarity of data presentation, the authors addressed all my comments. I am very happy to see this work!

=> Many thanks for your constructive comments (notably your earlier comments on RPP5) and your enthusiasm about our work!

In my opinion, the authors also addressed comments from the second reviewer including focusing the article, providing additional supplemental data, clarifying the figures and adjusting the discussion.

=> Thank you for some extra work (and time) accepting to take over on Reviewer 2.

Reviewer #3 (Remarks to the Author):

The authors revisions have clarified some of the key datasets and results and have improved the presentation of the results. These revisions have satisfied that concerns that I had on the first draft of the manuscript.

=> We also thank you for helping us clarifying our data and for your time.

Marie Mirouze, on behalf of all co-authors.